# Head and Neck Low Grade Chondrosarcoma—A Rare Entity

**DOI:** 10.3390/diagnostics13193026

**Published:** 2023-09-22

**Authors:** Camil Ciprian Mireștean, Cristiana Eugenia Simionescu, Roxana Irina Iancu, Mihai Cosmin Stan, Dragoș Petru Teodor Iancu, Florinel Bădulescu

**Affiliations:** 1Department of Medical Oncology and Radiotherapy, University of Medicine and Pharmacy Craiova, 200349 Craiova, Romania; 2Department of Surgery, Railways Clinical Hospital Iasi, 700506 Iasi, Romania; 3Department of Pathology, University of Medicine and Pharmacy Craiova, 200349 Craiova, Romania; cristiana.simionescu@umfcv.ro; 4Department of Pathology, Clinical Emergency County Hospital, 200642 Craiova, Romania; 5Oral Pathology Department, “Grigore T. Popa” University of Medicine and Pharmacy, 700115 Iași, Romania; 6Clinical Laboratory Department, “Sf. Spiridon” Emergency University Hospital, 700111 Iaşi, Romania; 7Department of Medical Oncology, Emergency County Hospital Vâlcea, 200300 Râmnicu Vâlcea, Romania; 8Department of Medical Oncology and Radiotherapy, “Grigore T. Popa” University of Medicine and Pharmacy, 700115 Iași, Romania; 9Department of Radiation Oncology, Regional Institute of Oncology, 700483 Iași, Romania

**Keywords:** low grade chondrosarcoma, head and neck cancers, radiotherapy, chemotherapy

## Abstract

Chondrosarcoma represents approximately 0.1% of all neoplasms of the head and neck and is considered a rare disease with a relatively good prognosis. The 5-year overall survival (OS) rate is estimated at 70–80%, being considered a disease with a low growth rate. Approximately 13% of all cases of chondrosarcoma are located in the region of the head and neck. We present the case of a 30-year-old patient without a medical history who reported dysphagia, swallowing difficulty, neck mass sensation and dysphonia that started insidiously after an upper respiratory tract infection. Subsequently, the patient was diagnosed with a low-grade glosso-epiglottic region chondrosarcoma and was multimodally treated with surgery followed by chemotherapy and radiotherapy. The radiation treatment was delivered with a Rokus M40 former Soviet Union cobalt machine without any image guidance capabilities. The inability to obtain resection margin information justified an aggressive adjuvant treatment with chemotherapy and radiotherapy. The early loss from the oncological record without recurrence of the disease could be associated in this case with the consequence of a major complication, of which we could assume an aspiration pneumonia secondary to a dysphagia associated with an aggressive multidisciplinary treatment. Large tumor size and positive resection margins (R1 resection) are risk factors that support an intensive adjuvant approach in order to reduce the risk of recurrence, but the low grade of tumor associated with a lower risk of recurrence as well as the adverse events (AE) of adjuvant radiotherapy and chemotherapy justify a more reserved therapeutic approach. Taking into account the longer life expectancy of these patients, it is recommended to use a more conformal irradiation technique in order to reduce doses to radiosensitive structures as well as to omit elective neck irradiation, taking into account the lower risk of lymph node involvement. The lack of guidelines, which include very rare tumors including low grade chondrosarcoma of the head and neck, makes a unified approach difficult, but the data presented in case reports could contribute to choosing the regimen that offers the best therapeutic ratio.

## 1. Introduction

Chondrosarcoma represents approximately 0.1% of all neoplasms of the head and neck and is considered a rare disease with a relatively good prognosis. The 5-year overall survival (OS) rate is estimated at 70–80%, being considered a disease with a low growth rate. A total of 1–3% of all cases of chondrosarcoma are located in the region of the head and neck. Chondrosarcoma has the highest incidence between the third and fourth decade of life, and cartilaginous neoplasms may have histological patterns that range from benign chondroid tumors to undifferentiated tumors [1].

WHO classifies chondrosarcomas in three degrees: cell density, differentiation and nucleus size are the factors considered in the classification; however, a simpler classification divides this type of tumor into two classes (low and high grade). Low-grade tumors usually have a favorable prognosis and are well differentiated, being associated with a low number of pleomorphisms and cellular atypia [2,3].

Low-grade tumors are usually well-differentiated with moderate cellularity and very few cellular pleomorphisms or atypia. All chondrosarcomas, regardless of grade, have histological features of hypercellularity, pleomorphism, mitotic activity, mixed intercellular matrix and cellular atypia. Surgical treatment is considered the backbone of the multidisciplinary approach, but the recurrence rate is relatively high for grade II and III tumors compared to low-grade tumors (80% vs. 17%) even after optimal surgical treatment. Although chondrosarcoma is considered a radio-resistant tumor, high-dose adjuvant irradiation is recommended in cases with positive surgical margins [2,4,5].

The purpose of this study is to present the case of a low-grade chondrosarcoma in a young patient with risk factors for recurrence (lack of data about an adequate resection margin and large tumor size) but also to emphasize the need to balance the adjuvant benefit of radiotherapy and chemotherapy with limiting adverse effects and the risk of dysphagia and aspiration in this category of tumors missed and orphaned by therapeutic guidelines.

## 2. Clinical Case

We present a 30-year-old patient without a significant medical history who was addressed to an ENT department in July 2012 with dysphagia, swallowing difficulty, neck mass sensation and dysphonia, the onset being insidious apparently following an episode of upper respiratory tract infection (URTI). The ENT examination revealed a mure-shaped polylobed tumor formation that displaced the suprahyoid epiglottis by mass effect and oriented it with the concavity towards the left hypopharyngeal wall, deformed it and left an equilateral triangular slit with a wall of approximately 5 cm. Hypo-pharyngoscopy and direct laryngoscopy identified a valecular cyst, which was subsequently removed, the evolution being favorable. The surgery was performed endoscopically with endotracheal intubation and general anesthesia.

In January 2013, the patient returned with inspiratory dyspnea, patient-reported dysphagia, stomatolalia and fever with progressive evolution. The ENT clinical examination highlighted a dyspneic patient with more pronounced inspiratory stridor, a multiform tumor process on the right valecula, which dislocated the suprahyoid epiglota on which it oriented with concavity towards the left hypopharyngeal lateral wall, deformed it and left an equilateral triangular slit with a side of approximately 5 cm. The chest X-ray did not show pathological pleuro-pulmonary aspects. The computer tomography (CT) exam revealed a polycystic-looking tumor formation with multiple septa inside and calcifications with 3–4 mm diameter that interest the right valecula, an aspect that suggested a possible chondrome. Under local anesthesia, tracheotomy was performed followed by tumor excision after hyoidectomy with a transvalecular approach. Epiglotopexy, laryngo-pharyngography, nasal-gastric feeding tube and Portex tracheal cannula were performed simultaneously. Postoperative evolution was marked by the development of a base of the tongue hematoma, surgical evacuation being performed. Favorable postoperative evolution was mentioned (at 4 days postoperatively, the tracheal cannula was suppressed and the patient was breathing normally).

The surgical pieces revealed a polylobate and polycyclic structure with a relatively smooth capsule and firm consistency through the transparency of the capsule brindle-like aspect red–violet areas, alternating with light areas. The tumor diameters were 105/65/85 mm. The section had a brindle-like, hard appearance of the whitish areas, and was the size of a rice grain, non-confluent, and separated by areas with a purple, fleshy, firm tint. The histopathological examination of the surgical pieces highlighted a nodular mycho-chondroid proliferation delimited by collagen fibrosis with reduced xanthomatous inflammation and chondrocytes. The cartilaginous tissue consisted of chondrocytes with irregular hyperchromic nuclei, rarely being binucleate, reduced mitotic activity and areas of bone metaplasia. In the periphery of the myxochondroid areas, striated muscle tissue was noticeable. The appearance of the histopathological examination suggested a hamartoma tumor (choristoma) that required a second opinion to differentiate from a low-grade chondrosarcoma (Figure 1, Figure 2, Figure 3 and Figure 4). Additional information about the resection margins was not specified. Immunohistochemistry confirmed the diagnosis of low-grade chondrosarcoma: p53 positive in the tumor cells and Ki67 positive in <5%.

In March 2013, the patient was addressed to the ENT department with swelling, hyperemia and pain at the lower pole of the cervical flap, patient-reported dysphagia and dehiscence of the wound on 1.5 cm with permanent bleeding. The ENT examination highlighted extensive valecular swelling in the suprahyoid epiglottis, the diagnosis being suppurative cervical cellulitis and acute epiglottitis. Subsequently, the patient was addressed to another ENT department, and a lingual tonsillectomy was performed. The histopathological examination revealed a fragment of the mucous membrane lined with squamous epithelium without atypical lymphoid tissue.

Five cycles of Epirubucine + Cisplatin were performed between April and July 2013; the patient then received external beam radiation using a Rokus M40 former Soviet Union cobalt machine without any image guidance capabilities in a total dose of 50 Gy/25 fractions/5 weeks/5 days per week between August and September. Without the tumor board concept being implemented at that time, radiotherapy was decided by the radiation oncologist as there was no information available about the resection margins. The patient was evaluated periodically in the oncology and ENT departments, presenting a performance index ECOG 2 (Eastern Cooperative Oncology Group) and dysphagia for solids and liquids with no significant loco-regional recurrence until March 2014, when he was lost from evidence.

## 3. Discussion

Representing approximately 11% of all primary malignant bone tumors and the second most common sarcoma subtype after osteosarcoma, chondrosarcoma of the head and neck region is rare. The base of the skull, the sinonasal tract, the jaw and the larynx are the anatomical locations reported to be associated with the occurrence of chondrosarcoma. Differential diagnosis with chondroblastic osteosarcoma and chondroid chordoma is considered difficult. With the 5-year overall survival (OS) rate estimated at 80%, surgery as a single method or followed by adjuvant radiotherapy is considered the optimal approach. For cases not eligible for surgery or that are associated with cosmetically mutilating effects, radiotherapy has been reported to be effective [6].

Horta et al. report a case of chondrosarcoma of the hyoid bone, a very rare subtype, with 20 cases being identified at that time in the literature. Worth noting in this case is the value of the computer tomography examination, which highlights the chondroid-like calcification originating from the hyoid bone, although previous cytology suggested pleomorphic adenoma. Prior to this case report, analyzing 19 cases of chondrosarcoma of the hyoid bone, Zhang et al. also mention the role of CT imaging and magnetic resonance imaging (MRI) in the diagnostic stage and of surgery followed by radiotherapy as the backbone treatment [7,8]. Retrospectively reviewing eight cases of chondrosarcoma of the head and neck region, the Yonsei University Medical College identified radiation therapy as delivered in 50% of cases as an adjuvant after surgery. The case in which complete excision could not be performed was associated with early death, and in three cases out of eight, relapse was diagnosed; the relapse being treated successfully in two cases [1,9].

Vučković et al. mention a case in which the patient was operated on 14 years after the diagnosis of chondrosarcoma, diagnosed at that time with a chondroma. The authors draw attention to the differential diagnosis of chondrosarcoma, a subtype of tumor characterized by mesenchymal transformation originating in hyaline cartilage but also to the possibility of transforming a chondroma into a low-grade chondrosarcoma. In all cases of head and neck submucosal tumors, chondrosarcoma should be considered as a differential diagnosis. After the left partial vertical laryngectomy treatment, no local recurrence or distant metastasis was reported in a 12-month follow-up. In a case of a right maxillary sinus tumor, grade III chondrosarcoma with skin and bone infiltration of the maxillary sinus and orbital floor was identified. Despite an aggressive adjuvant treatment, a higher tumor grade associated with extreme chromosomal instability was also associated with local recurrence. Sprekelsen et al. mention the tumor grade as being associated with the risk of recurrence and with the profile of therapeutic failure. In the case of low-grade tumors, local growth is predominant, and for high-grade head and neck chondrosarcoma, distant metastasis becomes the predominant pattern of failure, even after years from the initial treatment [2,10,11].

Clear cell chondrosarcoma, which usually involves the end of long bones, was reported in nine cases in the head and neck, including four cases of tumors of the larynx, the nasal septum of the jaw and the skull. Clear cell chondrosarcoma was associated with septal bulging, and the maxillary tumor did not involve the mucosa. In the case of laryngeal tumors, a tendency of recurrence was observed. Being a rare and slow-growing tumor, a long follow-up and a diagnosis based on immunohistochemical tests are necessary. An unusual onset through 12 days of vomiting was reported in the case of a low-grade right cricoid chondrosarcoma with intraluminal and extraluminal extension. Eighteen months after the resection of the trachea with end-to-end anastomosis, the patient was alive with no signs of recurrence. And in this case, a resection with margins free of tumor was considered sufficient with radiotherapy recommended to prevent local recurrence in the case of incomplete resection [12,13]. Rivero et al. mention an increased incidence in the 6th and 7th decade of life of laryngeal chondrosarcoma, being three times more frequent in men than in women. A total of 70–75% are located on the cricoid, and only 10–20% are located on the thyroid cartilage. More than 30 years ago, Burkey reported that the worst outcome was associated with chondrosarcoma of the paranasal sinuses [14,15].

Tejani et al. consider that tumor size and resection quality are the main risk factors associated with an unfavorable prognosis in head and neck sarcomas. A size of 5 cm is considered the cutoff value for the prognosis [16]. Although in most studies they consider the histological degree an essential prognostic factor, associating therapeutic failure and the increased rate of recurrences and metastases with higher degrees of sarcomas, the authors do not identify the histological tumor degree as a prognostic and predictive factor. Adjuvant radiotherapy is usually administered for cases with high histological grade chondrosarcoma and R1 resections at a dose between 50 Gy and 66 Gy in a standard fractionation regimen. The intensity modulated radiation therapy (IMRT) technique and the weekly low dose Cisplatin (30 mg/m^2^) and Ifosfamide-based chemotherapy are generally preferred for high-grade tumors. In the case of low-grade chondrosarcoma, the rate of distant metastases is relatively low (1.9%), but the analyzed group included only low-grade chondrosarcoma of the larynx [16,17]. Hoarseness and dyspnea but also the persistence and worsening of some symptoms associated with a submucosal lesion may suggest the presence of non-squamous laryngeal cancer, chondrosarcoma being one of the possible diagnoses [18].

Analyzing chondromas and chondrosarcomas of the base of the skull, Kremenevski et al. identify the incidence of chondrosarcomas as less than 0.15% of all intracranial tumors. The authors consider that chondrosarcoma has a better prognosis than chondroma, probably due to a more aggressive multimodal treatment. The overall survival (OS) for chondrosarcomas is 80% and 50% at 5 years and 10 years, respectively. Even if the authors note the benefit of adjuvant treatment with radiotherapy, chemotherapy is considered limited in value with maximum surgery being considered the basis of treatment. Although chondrosarcoma of the skull base is recognized by the high risk of recurrence, the multidisciplinary approach focuses on balancing the benefits/adverse effects to maintain the quality of life (QoL) [19].

A study that included 40 cases of chondrosarcomas of the head and neck treated surgically retrospectively evaluated the relationship between certain clinical and imaging characteristics and prognosis but also quantified the relationship between isocitrate dehydrogenase 1 and 2 (IDH 1,2) mutations and the evolution of the disease. The tumor infiltrated resection margin was associated with an unfavorable prognosis, but the IDH mutation did not influence the treatment results [20]. Lower rates of IDH mutation in laryngotracheal chondrosarcomas compared to skull base chondrosarcomas may suggest different mechanisms of tumorigenesis in the two anatomical sites mentioned [21].

A systematic review analyzing laryngeal chondrosarcoma cases identified 381 eligible cases. The most frequently involved anatomical structure was identified as the cricoid cartilage, and dyspnea, hoarseness and the sensation of mass in the throat were the most frequently reported symptoms. The surgical technique and the degree of histological differentiation were factors correlated with the prognosis [22]. In another systematic review that included only nine cases reported from 1946 to 2019, Jones and collaborators found a relatively low rate of loco-regional recurrences with only one of these cases recurring 4 years after surgical treatment. The authors also mention a relatively low rate of adjuvant treatment (two out of nine cases) and no case of distant metastasis. The study demonstrates the value of surgical treatment as the basis of the curative approach of head and neck chondrosarcoma. Both wound dehiscence and hematoma are major complications that require surgery, being reported with an incidence of 5.9% and 4.2%, respectively. Among the risk factors associated with these complications, neck dissection and neo-adjuvant chemo-radiotherapy are considered the most important [23,24,25]. A summary of the main articles on head and neck chondrosarcoma is included in Table 1. We present the objective and type of study, results and conclusions [6,7,8,9,10,11,12,13,14,15,18,24].

Almost 20 years ago, Eisbruch analyzed the anatomical strictures involved in late dysphagia/aspiration (DARS) in the case of HNC patients treated with chemotherapy based on Gemcitabine and high doses of concurrent radiotherapy for curative purposes, Coreland evaluation of swallowing and structural changes, videofluoroscopy, direct endoscopy and CT with radiotherapy plans obtained with the 3D-conformal technique (3D-CRT), intensity modulated radiotherapy (IMRT) and IMRT optimized for dysphagia reduction. A Cisplatin-based chemotherapy regimen was also proposed to evaluate the specificity of the changes associated with the treatment compared to a certain chemotherapy regimen. Prolongation of pharyngeal transit time, posterior tongue base weakness, lack of coordination between swallowing phases and limited laryngeal elevation, reduction in laryngeal closure and epigotic inversion were identified in the case of both regimens and were associated with a high aspiration rate. The study identifies the laryngeal constrictors, the glottic and supragotic larynx, as risk organs associated with dysphagia after radio-chemotherapy treatment. The authors demonstrated a moderate benefit of the IMRT technique in limiting the risk of late dysphagia, but by optimizing the IMRT plans related to dysphagia and reducing the radiation doses received by these structures, an additional benefit can be brought in limiting toxicities. Petkar et al. exploit this concept in a phase III randomized trial and evaluate whether sparing pharyngeal constrictor muscle and supraglottic larynx translates into the limitation of late dysphagia. The DARS trial (CRUK/14/014) assessed dysphagia according to the MD Anderson Dysphagia Inventory (MDADI) scale 12 months after treatment. The trial divides the cases of T1-4, N0-3 and M0 pharyngeal cancers into two equal groups that are treated with chemo-radiotherapy, using the IMRT technique for the delivery of radiotherapy and the IMRT technique optimized to limit dysphagia [26,27,28].

Dysphagia is one of the most serious tardive complications associated with an aggressive multimodal treatment including radiation therapy and chemotherapy, being a predictor for the risk of aspiration. Oropharynx and hypo-pharynx cancers but also the stage and size of the tumor are predictive factors of the risk of aspiration. Elderly patients and concurrent chemo-radiotherapy are also associated with an increased risk of late dysphagia and aspiration. Madan and collaborators report aspiration pneumonia as the cause for a 60% death rate in a group of 85 patients with head and neck cancers followed for a period of 12 years. The radio-biological model proposed by Christianen et al. identify the radiation dose received by the pharyngeal constrictor muscles and the supraglottic larynx as predictors for late severe dysphagia. The authors mention the impossibility of using a single dose-volume correlation to predict dysphagia. In the case of liquids, if the mean dose received by the supraglottic larynx is predictive, in the case of dysphagia for soft foods, both the radiation dose received by the superior pharyngeal constrictor muscle, the supraglottic larynx, and the age with a cutoff value of 65 years are associated with late toxicity. In this context, the use of a modern radiotherapy technique that reduces the doses received by these anatomical structures can significantly reduce the risk of late dysphagia and, as a consequence, reduce the risk for aspiration pneumonia [29,30,31,32]. Normal tissue complication probability (NTCP) models based on a specific end-point related to the risk of severe toxicity can be the basis of a decision algorithm between delivering the radiation dose treatment to the patient with the IMRT technique or proton beam therapy [33].

Although it is one of the most serious side-effects of HNC treatment, even associated with the risk of mortality, aspiration pneumonia associated with chemoradiotherapy and bioradiotherapy with Cetuximab is little investigated. Analyzing retrospectively the data obtained from 374 patients, of which 95 cases (25.4%) had associated aspiration pneumonia, the authors identified a correlation between aspiration pneumonia with treatment response and overall survival. Hospital treatment, hypoalbuminemia before treatment, N classification and oral hygiene were factors associated with the risk of aspiration pneumonia. Neck dissection and accelerated fractionation regimens are also considered risk factors for aspiration [34,35].

After an analysis of data from 3513 patients with HNC, aspiration pneumonia was identified in the history in 801 cases 5 months after the initiation of treatment. Hypopharyngeal tumors, male sex, advanced age, comorbidities and lack of surgical treatment before radio-chemotherapy were identified as risk factors for aspiration pneumonia in multivariate analysis. The hospitalization rate for patients with aspiration pneumonia is considered to be 84%, of which 45% are admitted to intensive care units. With a thirty-day mortality of 32.5% and a general risk of death of 45%, aspiration pneumonia must be considered as a possible late complication after radio-chemotherapy, especially in elderly patients. Xu et al. mention that 5 years after the end of the treatment, a quarter of this category of patients will be affected by aspiration pneumonia. Tumor stage is identified as a risk factor by Liu et al.; patients with ≥2 risk factors have a risk of 2.5 times more to aspire after the completion of radiotherapy treatment [36,37].

## 4. Conclusions

The early loss from the oncological follow-up programs of patients with a low degree of aggression tumor without evidence of a recurrence or metastasis can be the consequence of a major complication of which we can assume an aspiration pneumonia secondary to a dysphagia associated with an aggressive multidisciplinary treatment. Large tumor size and positive resection margins (R1 resection) are risk factors that support an intensive adjuvant approach to reduce the risk of recurrence, but the low degree of tumor, associating a lower risk of recurrence, as well as the adverse effects of adjuvant radiotherapy and chemotherapy are arguments that argue for an intensification of therapy. Given the long life expectancy of these patients, it is recommended to use an irradiation technique to reduce doses to radiosensitive structures as well as to omit elective neck irradiation, taking into account the reduced risk of lymph node involvement. The lack of guidelines that include very rare tumors, including low-grade chondrosarcoma of the head and neck, make a unified approach difficult, but the data presented in case reports could contribute to choosing the regimen that offers the best therapeutic ratio.

## Figures and Tables

**Figure 1 diagnostics-13-03026-f001:**
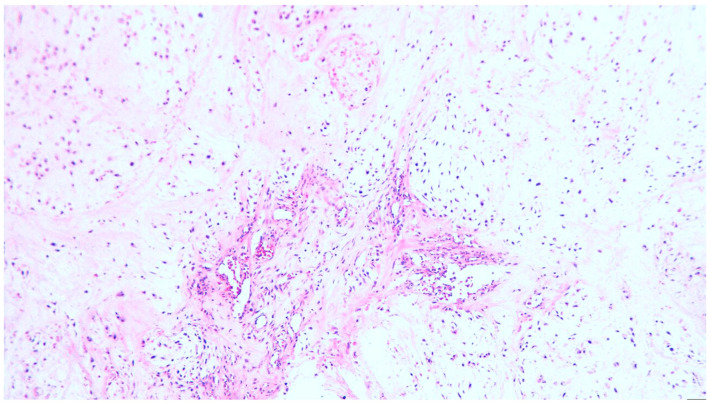
Chondrosarcoma grade I with lobular growth pattern, lobules separated by fibrous bands containing small blood vessels, hematoxylin and eosin (H&E) staining, ×100 magnification.

**Figure 2 diagnostics-13-03026-f002:**
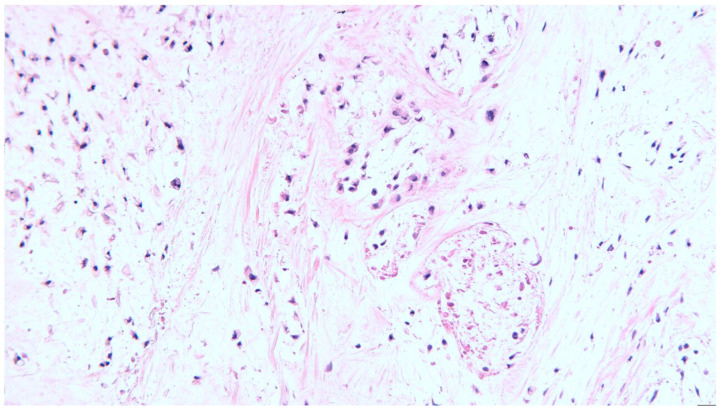
Chondrosarcoma grade I with lobular growth pattern, lobules separated by fibrous bands containing small blood vessels, H&E staining, ×200 magnification.

**Figure 3 diagnostics-13-03026-f003:**
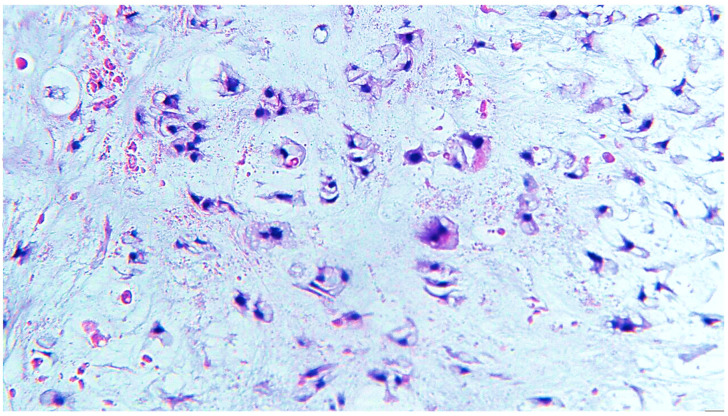
Chondrosarcoma grade I tumor cells with moderate eosinophilic or vacuolated cytoplasm and small, uniform nuclei, arranged in a myxoid matrix, H&E staining, ×400 magnification.

**Figure 4 diagnostics-13-03026-f004:**
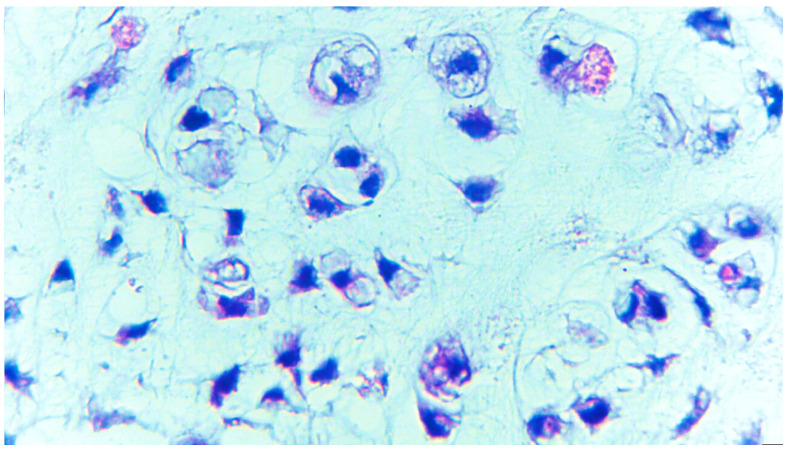
Chondrosarcoma grade I tumor cells with moderate eosinophilic or vacuolated cytoplasm and small and uniform nuclei with condensed chromatin, sometimes binucleate, ×400 magnification.

**Table 1 diagnostics-13-03026-t001:** A summary of the main articles on head and neck chondrosarcoma.

Main Objective	Article Type	Number of Cases	Results/Conclusions	Reference
To investigate clinical characteristics and treatment outcome of head and neck chondrosarcoma	Case series	8	Primary tumor sites were sinus, mastoid, jugular foramen and thyroid cartilage. A total of 50% of cases benefited from adjuvant radiotherapy. Three cases relapsed locally and two were treated for recurrence. One case operated with positive margins was associated with early death during the follow-up period	Lee et al., 2005 [1]
Description of a case of head and neck chondrosarcoma with aggressive behavior	Case report	1	A large number of molecular abnormalities and chromosomal instability is described in a case of grade III invasive chondrosarcoma of the maxillary sinus that recurs after adjuvant treatment.	Quevedo et al., 2007 [2]
To summarize diagnostic, pathological, clinical and evolutionary data of head and neck chondrosarcoma.	Review	Not specified	Conservative surgery should be the option for low-grade tumors; adjuvant radiotherapy should be considered in cases with higher local recurrence risk. OS is usually considered favorable	Coca-Pelaz A. et al., 2014 [6]
Description of an atypical case of hyoid bone chondrosarcoma	Case report	1	Painless palpable lump with growth of approximately 10 months. Considered extremely rare with only 20 cases in the literature.	Horta et al., 2015 [7]
Case report and review of current management options of a hyoid bone chondrosarcoma	Case report	1	Complete surgical removal is the recommended treatment, but radiotherapy can be used as adjuvant treatment.	Zhang et al., 2014 [8]
Description of chondrosarcoma of the larynx	Case report	1	Chondrosarcoma of the larynx should be considered as a differential diagnosis with chondroma for laryngeal submucosal tumors. The possible malignant transformation of the chondroma should also be taken into account.	Vučković et al., 2021 [10]
Clear cell chondrosarcoma of head and neck therapeutic diagnosis and outcomes presentation as a very rare entity.	Review	Not specified	Nasal tumor is associated with ballooning of the septum, and maxillary tumor does not involve the mucosa. Chondroblastic osteosarcoma differential diagnosis is essential for jaw tumors. Laryngeal localization tends to recur.	Mokhtari et al., 2012 [12]
Presentation of a case of low-grade chondrosarcoma of the cricoid cartilage with atypical onset 12-day history of vomiting	Case report and review of the literature	1	Tracheal cartilaginous ring can be associated with intraluminal and extraluminal extension. This subtype is rare and has a benign behavior. If the resection is incomplete, there is an increased risk of recurrence. In this case, adjuvant radiotherapy is necessary.	Gao et al., 2017 [13]
Report of a case of laryngeal chondrosarcoma located on thyroid cartilage	Case report	1	Chondrosarcoma of the larynx appears especially in the 6th–7th decades of life. The majority (70–75%) are located at the level of the cricoid and at the level of the thyroid cartilage (10–20%).	Pino Rivero et al., 2006 [14]
Retrospectively analyzes paranasal sinuses, mandible, temporal bone and larynx chondrosarcoma	Retrospective review	40 cases	The OS is 70% with a median follow-up of 3.5 years. Sinonasal chondrosarcoma is associated with the worst prognosis.	Burkey et al., 1990 [15]
Presentation of a case of laryngeal chondrosarcoma with a good prognosis.	Case report	1	Surrey with preservation of the larynx was associated with a favorable response and absence of recurrence 5 years after treatment.	Elktaibi et al., 2019 [17]
Evaluation of clinical data, imaging, histopathology and management outcomes for head and neck juxtacortical chondrosarcoma (HNJCS)	Case report and systematic review	9	One out of nine cases relapsed locally 4 years after the initial treatment. Most cases were low-grade tumors. Adjuvant treatment was delivered in 4 out of 9 cases.	Jones at al., 2019 [18]
Analysis of the correlations of the degree of histological differentiation with tumor parameters	Review	23 cases	Histological grade is correlated with survival; grade II and III tumors are more extensive and grow rapidly; there is no correlation between tumor size and histological grade.	Finn et al., 1984 [19]

## Data Availability

Not applicable.

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
