# Peer review of "Head and Neck Low Grade Chondrosarcoma—A Rare Entity"

_diagnostics, 2023, doi:10.3390/diagnostics13193026_

Round 1

Reviewer 1 Report

well written paper o a rare entity in Head and Neck, I think that the present work is suitable for publication

Author Response

Dear reviewer,

We express our gratitude for taking the time to evaluate the article and for the proposed critical comments. All additions made to the text at the suggestion of the other reviewershave been marked in blue.

We hope you will like this version of the manuscript and look forward to new possible suggestions.

Kind regards,

Camil Mirestean 

Reviewer 2 Report

well done and interesting paper to  read  and to know about 

Author Response

(The authors gave the same response as above.)

Reviewer 3 Report

The authors described a case of chondrosarcoma of the hypopharynx.

This is a rare entity. However, the manuscript needs many improvements.

The abstract seems a summary of the literature and do not report anything about this case report. Tha abstract must be rewritten.

At the end of the Introduction section the authors must report the aim of this paper.

Was pathological exam performed after the removal of vallecular cyst in 2012?

Radiological images must be added.

Was the tumor removed under general or local anesthesia? Was surgery performed through and open external approach?

Please report surgical margins.

Description of second pathological examination is too difficult to read.

More data about post-operative swallowing should be added. How was dysphagia evaluated?

Moderate editing of English language required

Author Response

Dear reviewer,

We express our gratitude for taking the time to evaluate the article and for the proposed critical comments.

The abstract was supplemented with relevant data about the case.

The purpose of the study was inserted at the end of the introduction.

It was stated that the histoplatological examination was performed after surgery. The type of anesthesia and surgical intervention was also specified. I have also summarized the histopathological result which does not highlight the presence of the tumor and whose details are not essential. We also specified that dysphagia was evaluated only subjectively, being reported by the patient. Regarding the resection margins, I cannot yet provide data until the end of August (the case pathologist is out of office). All additions made to the text have been marked in blue.

We hope you will like this version of the manuscript and look forward to new possible suggestions

Kind regards,

Camil Mirestean 

Reviewer 4 Report

Please provide bibliography for this paragraph -Chondrosarcoma, represents approximately 0.1% of all neoplasms of the head and neck, being considered a rare disease but with a relatively good prognosis. The 5-year overall survival (OS) rate is estimated at 70-80%, being considered a disease with a low growth rate. 1–3% of all cases of chondrosarcoma, are located in the region of the head and neck. Chondrosarcoma has the highest incidence between the third and fourth decade of life and cartilaginous neoplasms may have histological patterns ranging from benign chondroid tumors to undifferentiated un-differentiated tumors

The authors could insert also the CT scan.

i suggest the authors regarding the case presentation to be more structured.

In the discussion chapter to include more recent published articles.

Moderate revision.

Author Response

Dear reviewer,

We express our gratitude for taking the time to evaluate the article and for the proposed critical comments. To begin with, we added the reference for the initial paragraph. In the case of the presentation, we added details about the operative technique and mentioned that the dysphagia was only patent related. We have also summarized the 2nd pathological result which, not revealing the presence of the tumor after treatment, was not relevant to be presented in detail. We have also regrouped in the presentation of the case pathological results with immunohistochemistry and surgery with the postoperative evolution.

Unfortunately, at the time the case was treated (2012), CT imaging was often requested in other private centers due to very limited access and waiting lists. In this context, we only have access to the result written in the archive of the oncology clinic. The discussion chapter was also completed with new references (but the absence of recent case reports in the literature should be mentioned) and the abstract included more data about the case. Regarding the resection margins, I cannot yet provide data until the end of August (the case pathologist is out of office). All additions made to the text have been marked in blue.

We hope you will like this version of the manuscript and look forward to new possible suggestions.

Kind regards,

Camil Mirestean 

Round 2

Reviewer 3 Report

The manuscript has been improved. However, some issues remain.

The authors should specify if pathological exam was performed after the removal of vallecular cyst in 2012.

Radiological images and data about surgical margins must be added.

Author Response

Dear Reviewer,

Thank you for evaluating the manuscript and the proposed recommendations. We specified that it is the histoplatological examination of the surgical piece, without a previous biopsy and technically the resection limits could not be established. For this reason, adjuvant radiotherapy was also delivered, the overtreatment option being preferred and to avoid the risk of suboptimal treatment. Unfortunately, the technical and accessibility limitations of computer tomography imaging 10 years ago and the acquisition of computer tomography examinations in other centers outside the university clinic, but also the non-existence of tumor board with comprehensive pre-treatment analysis of the case in a multidisciplinary team, but also the use of a very outdated irradiation technique based only on anatomical landmarks are factors that limited the imaging analysis of the case. Probably, more than the rarity of the case, the argument of the presentation was the "lesson to be learned from history" with a special focus on the need to evaluate rare cases of head and neck cancers in multidisciplinary committees and the importance of quality radiotherapy to reduce adverse effects, in especially dysphagia and aspiration in cases with a favorable prognosis. I mentioned that the additions to the text were marked in dark blue.

Kind regards,

Camil Mirestean

Round 3

Reviewer 3 Report

Thanks for your response.